# Mapping Impervious Surface Using Phenology-Integrated and Fisher Transformed Linear Spectral Mixture Analysis

Linke Ouyang [1,†], Caiyan Wu [2,†], Junxiang Li [2], Yuhan Liu [1], Meng Wang [1], Ji Han [1,*], Conghe Song [3], Qian Yu [4] and Dagmar Haase [5,6]

1   Shanghai Key Laboratory of Urbanization Processes and Ecological Restoration, School of Ecological and Environmental Sciences, East China Normal University, Shanghai 200241, China; 52193903005@stu.ecnu.edu.cn (L.O.); 52193903007@stu.ecnu.edu.cn (Y.L.); wangmeng00123@163.com (M.W.)
2   Department of Landscape Architecture, School of Design, Shanghai Jiao Tong University, Shanghai 200240, China; caiyanwu@sjtu.edu.cn (C.W.); junxiangli@sjtu.edu.cn (J.L.)
3   Department of Geography, University of North Carolina at Chapel Hill, Chapel Hill, NC 27599, USA; csong@email.unc.edu
4   Department of Geosciences, University of Massachusetts, Amherst, MA 01003, USA; qyu@geo.umass.edu
5   Department of Geography, Humboldt-Universität zu Berlin, 10117 Berlin, Germany; dagmar.haase@ufz.de
6   Department of Computational Landscape Ecology, Helmholtz Centre for Environmental Research-UFZ, 04318 Leipzig, Germany
*   Correspondence: jhan@re.ecnu.edu.cn
†   These authors contributed equally to this work.

**Abstract:** The impervious surface area (ISA) is a key indicator of urbanization, which brings out serious adverse environmental and ecological consequences. The ISA is often estimated from remotely sensed data via spectral mixture analysis (SMA). However, accurate extraction of ISA using SMA is compromised by two major factors, endmember spectral variability and plant phenology. This study developed a novel approach that incorporates phenology with Fisher transformation into a conventional linear spectral mixture analysis (PF-LSMA) to address these challenges. Four endmembers, high albedo, low albedo, evergreen vegetation, and seasonally exposed soil (H-L-EV-SS) were identified for PF-LSMA, considering the phenological characteristic of Shanghai. Our study demonstrated that the PF-LSMA effectively reduced the within-endmember spectral signature variation and accounted for the endmember phenology effects, and thus well-discriminated impervious surface from seasonally exposed soil, enhancing the accuracy of ISA extraction. The ISA fraction map produced by PF-LSMA (RMSE = 0.1112) outperforms the single-date image Fisher transformed unmixing method (F-LSMA) (RMSE = 0.1327) and the other existing major global ISA products. The PF-LSMA was implemented on the Google Earth Engine platform and thus can be easily adapted to extract ISA in other places with similar climate conditions.

**Keywords:** impervious surface area; phenology information; Fisher transformation; linear spectral mixture analysis; endmember variability; Google Earth Engine; seasonally exposed soil; VIS model; Shanghai; Landsat

## 1. Introduction

More than 55% of the world's population lived in urban areas by the end of 2018, and the proportion is projected to reach 68% by 2050 [1]. Rapid urbanization resulted in a marked increase in impervious surface area (ISA) at local [2,3], regional [4], and global scales [5] in the past few decades. The impervious surface area (ISA) refers to the artificial land surface covered by water-resistant materials, such as residential, public facility, industry, traffic, and the in-construction land [6]. The increase in ISA has brought out serious adverse environmental and ecological consequences, such as urban heat island intensification [7], habitat loss [8], urban flooding [9,10], and public health impacts [11]. Therefore, the ISA emerged as an important environmental indicator [6], which plays a

critical role in sustainable urban development [12]. How to extract ISA accurately from remotely sensed data is essential for urban ecology and planning.

Among the methods extracting ISA from remotely sensed data, the spectral mixture analysis (SMA) [13] is widely used because of its ability to deal with the variation at the subpixel level. SMA is a physically-based image-analysis approach that extracts fractional proportions of fundamental components, called endmembers, at the subpixel level [14]. Due to the ease of access and long-term data archive, Landsat images have been widely employed to monitor the spatio-temporal dynamics of ISA [15–17]. The SMA hypothesizes that the spectral signature of a mixed pixel is a combination of the spectral signatures of its endmembers weighted by their areal fraction at the subpixel level [14]. The SMA includes the linear spectral mixture analysis (LSMA) and the nonlinear spectral mixture analysis. The former only considers the single scattering of the land surface, while the latter considers the multiple-scattering caused by the 3D structure of the land surface [18]. Although nonlinear SMA is superior when implemented in urban areas with vertical structures [19], the LSMA is still widely used because of its simplicity and low sensitivity to noise. The V-I-S model proposed by Ridd [20] is a well-known urban LSMA model, which assumes that an urban pixel is formed of three endmembers: vegetation, impervious surface, and soil. The V-I-S model has been widely used to extract the three components at the subpixel level, especially the impervious surface [21–25]. However, the high landscape heterogeneity in urban areas influences the accuracy of LSMA due to the spatial-temporal variability of endmember signatures [26–29].

Many studies have addressed the effects of endmember spectral variations to improve the accuracy of SMA [14,28,29]; it was reviewed and summarized into five approaches [26]. The iterative mixture analysis, e.g., multiple endmember spectral mixture analysis (MESMA) [14], was proposed to address the problem of endmember signature variation. However, the overfitting of this model hinders the accuracy of the unmixing, especially when the endmember spectral signature similarity of soil and impervious surface is not negligible. The spatially adaptive SMA technique (SASMA) was developed to automatically extract and synthesize the "most representative" endmembers adaptively in space, account for spatial variation in endmember spectral signatures [29]. In particular, spatially adaptive endmembers were adopted to take into account spatial variation in endmember signatures. However, the SASMA does not consider the between-endmember spectral similarity. The endmember spectral feature selection, e.g., Stable Zone Unmixing (SZU) [30], was designed for SMA with hyperspectral imagery. The spectral weighting, e.g., normalization [24] and derivation [31], can reduce within-endmember spectral variations. However, it also leads to the between-endmember spectral similarity. The spectral modeling needs prior knowledge and preset parameters, which introduce errors [32]. Spectral transformation is an important process before unmixing to reduce the dimension of input endmembers' features. There are several spectral transformation methods, such as the principal component (PC) transformation [22,24,29], maximum noise fraction (MNF) transformation [25,33], and Fisher transformation [34], which are used to process the remotely sensed data before SMA. Among these transformations, the Fisher transformation can improve the accuracy of the unmixing since it can enhance the between-endmember spectral variability and reduce the within-endmember spectral variability [35]. However, the inherent spectral similarity between the soil and impervious surface still hinders the accurate ISA extraction. Therefore, we still need innovative approaches to further enhance the discrimination between the soil and impervious surface.

Phenology, another important dimension of information, has the potential to distinguish the impervious surface from seasonally exposed soil [33]. The urban landscape is complex, constantly undergoing changes. Some of these changes are permanent alterations from the natural landscape to the impervious surface, while others are the seasonal variation caused by phenology, such as the crop harvest and rotation, tree defoliation, and lawn dormancy. Several studies have demonstrated the utility of phenology in improving ISA extraction. Small [36] proposed a seasonal stability index to represent phenological

information, which calculates the spectral stability of each pixel's fraction of the S-V-D model for impervious surface (S), vegetation (V), and shadows (D). However, the seasonal stability index calculation can introduce errors during the unmixing process with the S-V-D model due to the collinearity among the three components. Yang, et al. [37] proposed a temporal mixture analysis (TMA) technique that fully utilizes the phenological information by using the rearranged MODIS NDVI time-series datasets at the stable temporal zone, i.e., the highest six NDVI values through a year, to reduce the effect of endmember variability. Li and Wu [38] proposed a phenology-based TMA that incorporates the entire year of NDVI to enhance the ISA estimation accuracy. The TMA shows as an efficient way to incorporate phenological information into the unmixing process. However, it still cannot effectively discriminate between ISA and bare soil, hence not suitable for the urban area with seasonally exposed soil [37]. Moreover, The TMA was initially designed for MODIS images, thus the algorithm is not suitable for low temporal but high spatial resolution satellite images, such as the Landsat images [39]. Another problem is that the landscape phenology profile is complex and hard to predict [40–42], making the NDVI temporal profile of the pixels difficult to be defined. Therefore, how to incorporate phenological information to enhance the accuracy of ISA remains a challenge.

This study aims to develop a novel method that incorporates phenological information and Fisher transformation into the LSMA (PF-LSMA) to improve the accuracy of ISA extraction. We hypothesize that directly integrating phenological information into SMA by compositing the spectral features over seasons can enhance the separation of seasonally exposed soil and impervious surface, consequently improving the accuracy of ISA extraction.

## 2. Materials and Methods

### 2.1. Study Area

Our study area is Shanghai, China (120°52′–122°12′ E, 30°40′–31°53′ N), which lies on the eastern estuary of the Yangtze River Delta, spanning a total area of 6340.5 km$^2$ with a total population of 24.15 million in 2015. Shanghai has experienced rapid urbanization in the past thirty years and has become the engine of regional economic growth in the Yangtze River Delta, a major urban agglomeration in China. The Gross Domestic Product (GDP) of Shanghai reached CNY 2.5 trillion (~USD 403.4 billion) in 2015 [43]. Rapid economic growth has dramatically converted natural and/or semi-natural landscapes into the impervious surface. The average growth rate of urban land is 49.5 km$^2$ per year, with the urbanization rate during 2005–2015 being nearly twice that of 1985–1995 [44].

Shanghai is located in the northern subtropical monsoon climate zone, where the native vegetation is dominated by evergreen broadleaved forests, deciduous forests, and mixed forests. Currently, the urban greenspaces are composed of different vegetation types, primarily planted evergreen/deciduous broad-leaved trees and lawns [45]. The northern subtropical monsoon climate also benefits agriculture with diverse crops. Cropland comprises approximately 30% of the land administered by the Shanghai municipality in 2015, which covers a total area of 1898 km$^2$ in the suburban region. The cropland is typically farmed with a double-cropping system [42,46].

### 2.2. Methodology

#### 2.2.1. Workflow

The proposed method in this study is called a phenology-integrated and Fisher transformed linear spectral mixture analysis (PF-LSMA). The PF-LSMA includes five steps: (1) image pre-processing, (2) phenological information integration, (3) endmember selection, (4) Fisher transformation, and (5) linear spectral mixture analysis (LSMA) in Fisher feature space (Figure 1). All steps were implemented in the Google Earth Engine (GEE) platform except for Fisher Discriminate analysis (FDA). The Google Earth Engine (GEE) code produced by this study is available in https://code.earthengine.google.com/54cb5465bba6

49176480a5219fcb0e0b, accessed on 20 March 2022. The R code for FDA is available in https://github.com/linkeouyang/Fisher-transformation.git, accessed on 20 March 2022.

The performance of PF-LSMA is tested against single-date image Fisher transformed LSMA (F-LSMA), i.e., without phenological information, to demonstrate the effect of phenology on ISA accuracy. Moreover, we also compared the ISA derived from PF-LSMA with existing global ISA products. Figure 1 shows the details of the workflow.

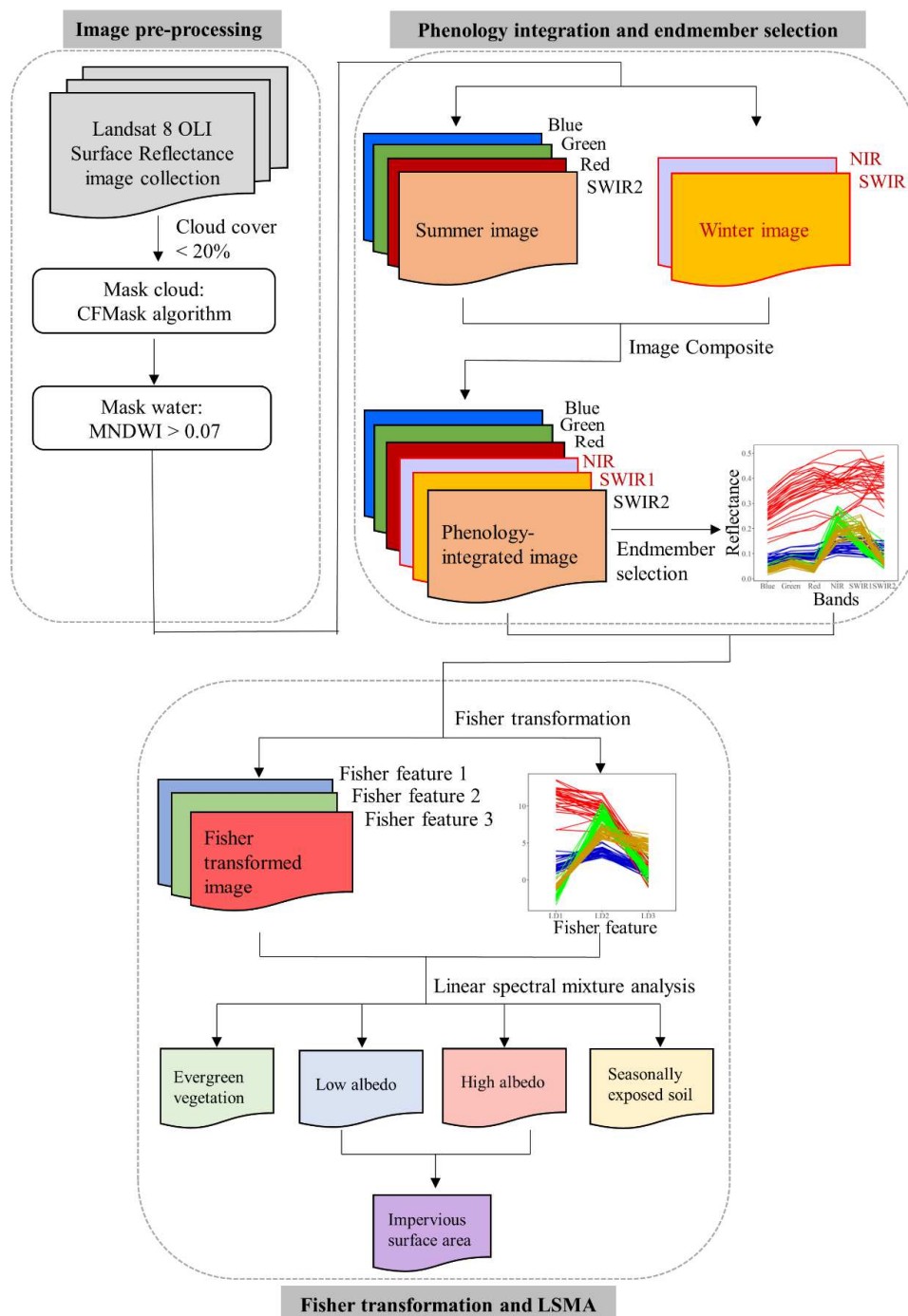

**Figure 1.** Flowchart of the phenology-integrated and Fisher transformed linear spectral mixture analysis (PF-LSMA).

2.2.2. Image Pre-Processing

Image pre-processing includes the masking of cloud, cloud shadow, and water. Four cloudless (<20%) Landsat 8 OLI images (Row/Path: 038/118, 039/118), acquired on 3 August 2015 and 27 February 2016, were used in this study. The images are USGS Landsat 8 Collection 1 Level 2 Surface Reflectance product.

To obtain valid pixels for unmixing, multi-layer masking was conducted to remove water, clouds, and cloud shadows. Clouds and cloud shadows were removed using the C Function of the Mask (CF Mask) algorithm [47] embedded in Google Earth Engine (GEE). Water was masked using the Modified Normalized Difference Water Index (MNDWI) proposed by Xu [48]:

$$MNDWI = (Green - MIR)/(Green + MIR) \tag{1}$$

where *Green* and *MIR* are the reflectance values of Landsat 8 OLI bands 3 and band 6, respectively. A threshold value of 0.07 was used to effectively mask the water surfaces [49,50]. All the abovementioned datasets are available in GEE (Table 1).

**Table 1.** USGS Landsat 8 images used in this study. The images are Collection 1 Level 2 surface reflectance data with terrain correction.

| Date | Image ID in GEE |
| --- | --- |
| 08/03/2015 | LANDSAT/LC08/C01/T1_SR/LC08_118038_20150803 |
| 08/03/2015 | LANDSAT/LC08/C01/T1_SR/LC08_118039_20150803 |
| 02/27/2016 | LANDSAT/LC08/C01/T1_SR/LC08_118038_20160227 |
| 02/27/2016 | LANDSAT/LC08/C01/T1_SR/LC08_118039_20160227 |

2.2.3. Endmember Model and Phenological Information Integration

The endmember model for the PF-LSMA consists of high albedo, low albedo, evergreen vegetation, and seasonally exposed soil (H-L-EV-SS). Due to the within-endmember and between-endmember variations in the spectral signature [26,27], appropriate endmember selection [29,51,52] is key to the accurate estimation of subpixel endmember fractions using SMA [23,53]. In this study, the H-L-EV-SS endmember model was developed based on the V-I-S model [20], which includes three main endmembers of vegetation, impervious surface, and soil. To address the spectral variability of impervious surface, the ISA is usually separated into two components of high albedo and low albedo, creating the V-H-L-S (vegetation-high albedo-low albedo-soil) model [24,25,33]. The high albedo refers to the impervious surface with high reflectance, including concrete, and light rooftops. The low albedo includes the impervious surface with low reflectance values, such as asphalt roads, dark roofs, and shadowed concrete. By integrating the spectral signature of different seasons, the composite spectra of vegetation and soil show distinguishable characteristics. Specifically, the vegetation and soil endmembers were modified to be evergreen vegetation and seasonally exposed soil in Shanghai, accounting for their phenological characteristics. The evergreen vegetation refers to areas covered by species that have no significant phenological change throughout the year. The seasonally exposed soil refers to areas mainly covered by crops or other deciduous vegetation during the growing season but exposed as bare soil during the other times of the year.

The phenology can be incorporated by composing the satellite images from two different seasons [54], i.e., the summer time (or the growing season) and winter time (or the dormant season). In this study, the image obtained on 3 August 2015 was used as the summer image, and the image acquired on 27 February 2016 was selected as the winter image. To maximize the differences in spectral signatures between endmembers, based on the observation of spectral characteristics of endmembers in summer and winter images (Figure 2f,h,j,l), we selected a total of six bands, including NIR band (0.851–0.879 μm) and SWIR1 band (1.566–1.651 μm) from the winter image, and the blue band (0.452–0.512 μm), green band (0.533–0.590 μm), red band (0.636–0.673 μm), and SWIR2 (2.107–2.294 μm) from

the summer image. The 6-band composite image where all endmembers have identifiable characteristics (Figure 2g,i,k,m) was used as input data layers in the subsequent processes.

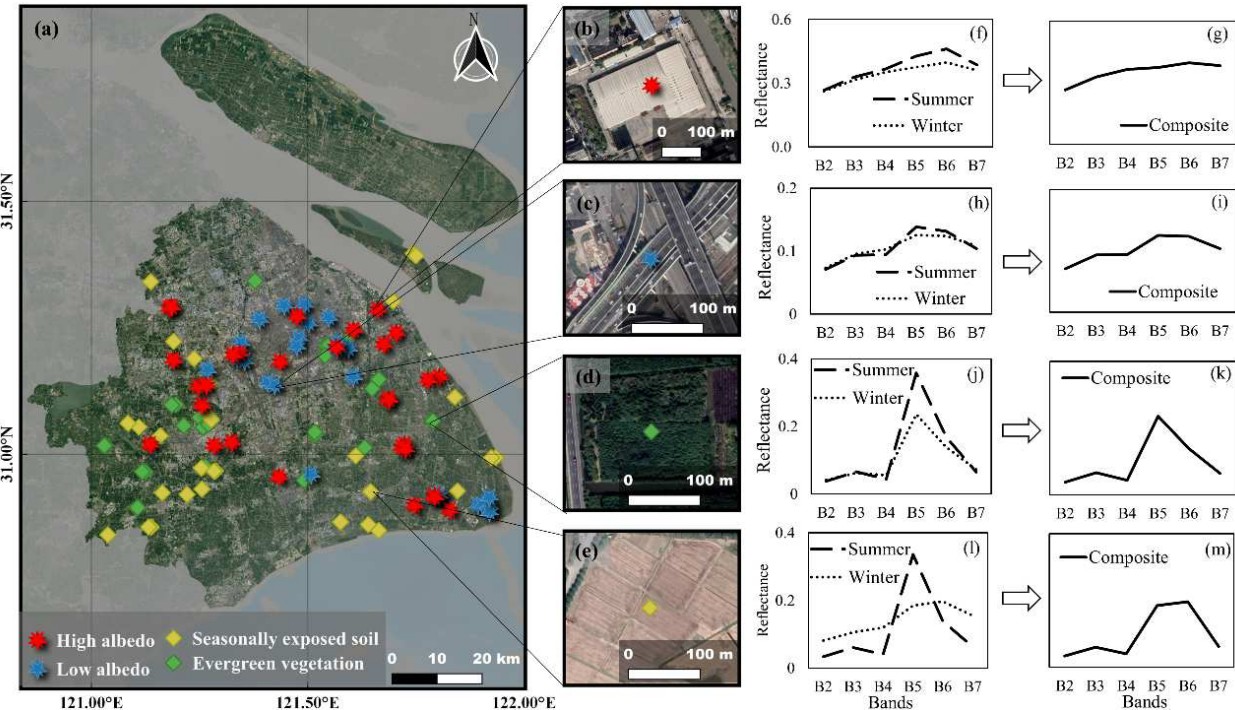

**Figure 2.** The spatial distribution of endmember pixels (**a**). The example location of high albedo (**b**), low albedo (**c**), evergreen vegetation (**d**), and seasonally exposed soil (**e**). The spectra in summer and winter image, and composite image of high albedo (**f**,**g**), low albedo (**h**,**i**), evergreen vegetation (**j**,**k**), and seasonally exposed soil (**l**,**m**).

### 2.2.4. Endmember Selection

Properly selecting endmembers is key to spectral mixture analysis [28,29]. There are several ways to select endmembers, such as images-based endmember selection [25,55], spectral library-based selection [35,56,57], and virtually generated endmembers [58]. Among them, the image-based endmember selection is often the preferred because (1) the end-members' spectra are obtained under the same imaging condition (atmosphere, viewing, and illumination angles); (2) they are readily available; and (3) there are lower computational errors and they are cost-effective compared with virtual endmembers. Therefore, we utilized the image-based approach with visual interpretation to select endmembers. The image-based endmembers are generally selected from representative homogeneous pixels from the vertex of spectral scatter plots [59]. In this study, we conducted the principal component analysis (PCA), then the first three components were used to generate a feature space scatterplot [22,24,29]. The pixels located at the vertex of the feature space were identified and evaluated in the original winter and summer images, and the pure pixels were visually identified as endmembers. The endmember candidates were selected based on the following rules: First, the evergreen vegetation endmember candidates (Figure 2d) are the pure pixels, which have consistent spectral signatures in all bands in both winter and summer images (Figure 2j,k). In this study, evergreen vegetation was mainly evergreen trees in urban parks. Second, the seasonally exposed soil endmember candidates (Figure 2e) are the pure pixels whose spectral characteristics show seasonal variations, i.e., the spectral characteristic of vegetation in the summer image and bare soil in the winter image (Figure 2l,m). It mainly locates in cropland. Third, the spectral signatures of high albedo and low albedo candidates show considerable seasonal stability in the winter and summer images. The high albedo endmember candidates were usually found at the rooftops

of the newly constructed buildings (Figure 2b,f,g), whereas the low albedo endmember candidates were mainly selected from the asphalt roads (Figure 2c,h,i). We finally picked out 30 pure pixel candidates for each type of the endmember (Figure 2a), and the spectral signatures were extracted from the composite image. The mean spectral signatures of all candidate pixels were used as the final spectral signature for each endmember [28].

### 2.2.5. Spectral Transformation Using Fisher Linear Discriminant Analysis

In this study, a spectral transformation, i.e., the Fisher transformation, was implemented to the summer-winter image composite to reduce the spectral variability within an endmember and maximize the spectral variability between endmembers. Instead of using original spectral data, the transformed Fisher features were used as inputs for the linear spectral mixture analysis (LSMA). Previous studies demonstrated that Fisher transformation can effectively enhance the accuracy of ISA estimation since it can strongly reduce the ratio of within-endmember to between-endmember variation when incorporated into LSMA [35,60].

Fisher transformation can be implemented in two steps: (1) calculating the weight matrix with the selected endmembers (Section 2.2.4) by Fisher discriminate analysis (FDA), and (2) transforming the original image's reflectance into the Fisher features using this weight matrix. To find the weight matrix $w$, the between-endmember scatter matrix $S_b$ and within-endmember scatter matrix $S_w$ [61] need to be calculated before defining the objective function $J$. The aim of the FDA is to find the $w$ which maximum the objective function $J$.

$$S_b = \frac{1}{N} \sum_{c \in C} n_c (\overline{x}_c - \overline{x})(\overline{x}_c - \overline{x})^T,$$
$$S_w = \frac{1}{N} \sum_{c \in C} \sum_{i=1}^{n_c} (x_{c,i} - \overline{x}_c)(x_{c,i} - \overline{x}_c)^T, \tag{2}$$
$$J = \frac{w^T S_b w}{w^T S_w w}$$

where $C$ is the assemble of endmember groups, and $c$ represents each endmember group; $N$ is the total number of endmember pixels, and $n_c$ is the number of pixels for group $c$; $\overline{x}$ is the mean of all endmembers, and $\overline{x}_c$ is the mean of group $c$; $x_{c,i}$ represents $i^{th}$ pixel in endmember group c. $w$ is the weight matrix.

The weight matrix $w$ is the eigenvector of the matrix $S_w{}^{-1}S_b$, and is solved by the following equation [35]:

$$S_w{}^{-1}S_b w = \lambda w \tag{3}$$

where $\lambda$ is a nonzero eigenvalue. In this study, three nonzero eigenvalues were used as three Fisher projection vectors ($w = [w_1, w_2, w_3]$), which was solved in $R$ software using the *MASS* package. Three features were used to avoid the issue of over-determination, which mainly occurs when the number of features selected for LSMA is more than the number of endmembers [26].

The reflectance of the composite image was then transformed into the Fisher feature space using the following equation:

$$F_x = [F_{w1}, F_{w2}, F_{w3}]^T = w \cdot [R_{B2}, R_{B3}, \ldots, R_{B7}]^T \tag{4}$$

where $F_x$ is the $3 \times 1$ matrix of transformed Fisher features for pixel $x$. $[R_{B2}, R_{B3}, \ldots, R_{B7}]^T$ is the $6 \times 1$ matrix of reflectance for pixel $x$. $w$ is the $3 \times 6$ weight matrix.

To check the validity of the H-L-EV-SS model, the scatterplot of the Fisher transformed composite image was generated in the Fisher feature space, and the endmembers were confirmed to be found in the vertex of the scatterplot (Figure 3).

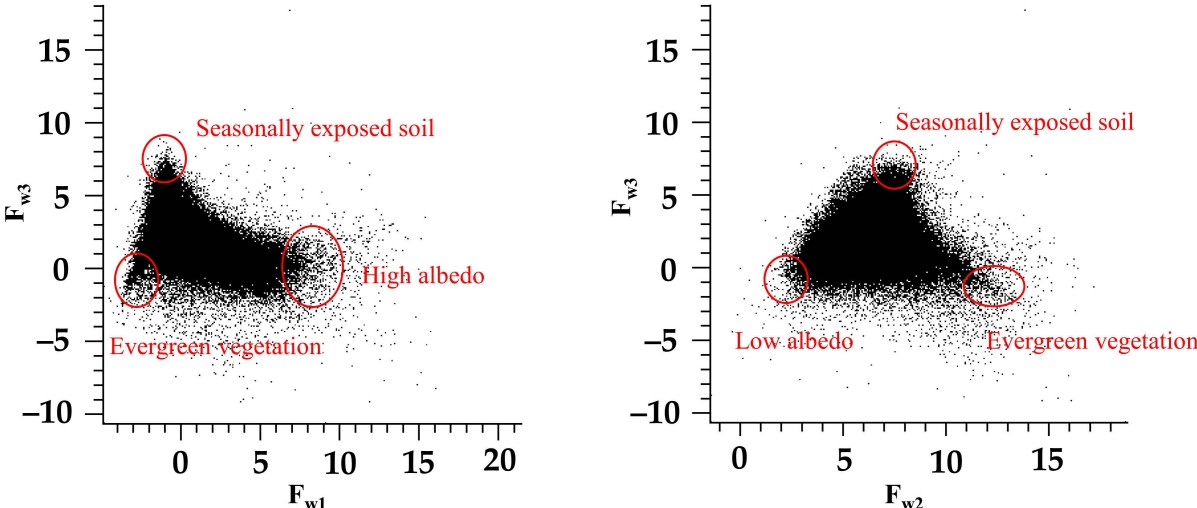

**Figure 3.** The scatterplot of Fisher transformed composite image.

### 2.2.6. Linear Spectral Mixture Analysis in Fisher Feature Space

The linear spectral mixture analysis (LSMA) is based on the assumption that the spectral features of a mixed pixel are the linear combination of the spectral signatures of the endmembers weighted by their areal proportion in the pixel [14]. Using the Fisher transformed features, the LSMA function can be written as:

$$F_x = F_h \cdot f_h + F_l \cdot f_l + F_{ev} \cdot f_{ev} + F_{ss} \cdot f_{ss} + e_b,$$
$$f_h + f_l + f_{ev} + f_{ss} = 1,$$
$$0 \le f_h, f_l, f_{ev}, f_{ss} \le 1$$
(5)

where $F_x$ is the $3 \times 1$ matrix of transformed Fisher features for pixel $x$. $F_h$, $F_l$, $F_{ev}$, $F_{ss}$ are the transformed Fisher features of high albedo, low albedo, evergreen vegetation, and seasonally exposed soil, respectively, and $f_h$, $f_l$, $f_{ev}$, $f_{ss}$ is the fraction of them. $E_b$ is the residual.

Moreover, the "sum-to-one" and "non-negativity" constraints are enforced. The "sum-to-one" constraint is a straightforward application of Lagrange multipliers, and "non-negativity" constraint is addressed using the active set method.

This step was achieved by the GEE algorithm using the Fisher transformed composite image and the mean Fisher features of each endmember as input data. After this step, the fraction maps of each endmember type were generated. In this study, ISA fraction is what we are interested in, and it was calculated by adding the fraction of high and low albedo together:

$$f_{ISA} = f_h + f_l,$$
(6)

where $f_h$ and $f_l$ are the fractions of high albedo and low albedo, respectively. $F_{ISA}$ is the fraction of the impervious surface.

### 2.2.7. Validation

The land use and land cover (LULC) data of 264 randomly distributed circles (1-km radius) in Shanghai were used as the reference dataset to evaluate the accuracy of the ISA fraction map (Figure 4). The LULC, which includes eight categories, was adopted from Li, et al. [62]. It was visually interpreted and digitalized from aerial photos with a spatial resolution of 0.5 m [45]. The ISA fraction in each circle was calculated from the proportion of the total area of the residential, public facility, industrial area, and transportation to the entire area of the circle.

This reference dataset was not only used to validate the ISA estimated by PF-LSMA, but also used to validate other ISA products, including the ISA fraction map generated

by single-date image Fisher transformed LSMA (F-LSMA) and two existing global ISA products as comparisons. The Landsat 8 image collected on 27 February 2016, which was used as the winter image to PF-LSMA, was used as the single-date image for F-LSMA. As discussed in Section 2.2.3, the seasonally exposed soil has no vegetation cover and exposed as bare soil in the winter image. Therefore, the endmembers model, including high albedo, low albedo, vegetation, and soil [25] was used for F-LSMA. The pixels where the endmembers selected for the PF-LSMA model were located as the image endmember were selected for F-LSMA. The same procedure of Fisher transformation was also conducted for F-LSMA using the single-date-image endmembers.

To further evaluate the quality of ISA derived from PF-LSMA, we compared it with the existing global impervious surface products, including the Normalized Urban Areas Composite Index (NUACI) [63] and the Global Artificial Impervious Areas (GAIA) [5]. They are pixel-based ISA products with 30 m spatial resolution, which were also produced using Landsat images.

For each ISA product, the ISA fraction of each validating circle was calculated using Equation (7):

$$f_{ISA,cir} = \sum_{x}^{n}(f_{ISA,x} \cdot S_x) \ / S_{cir} \tag{7}$$

where $f_{ISA,cir}$ is the ISA fraction of validation each circle. $S_{cir}$ is the total area of each circle. $n$ is the total amount of pixels in the circle, $S_x$ is the area of pixel $x$. For the ISA fraction map generated by PF-LSMA and F-LSMA, $f_{ISA,x}$ is the ISA fraction of pixel $x$. For NUACI and GAIA datasets, $f_{ISA,x}$ is 1.

The root mean squared error (RMSE) and the Pearson correlation coefficient (R) were calculated for each dataset. The RMSE measures the differences of ISA fraction between the estimated and the reference values within the validation circle.

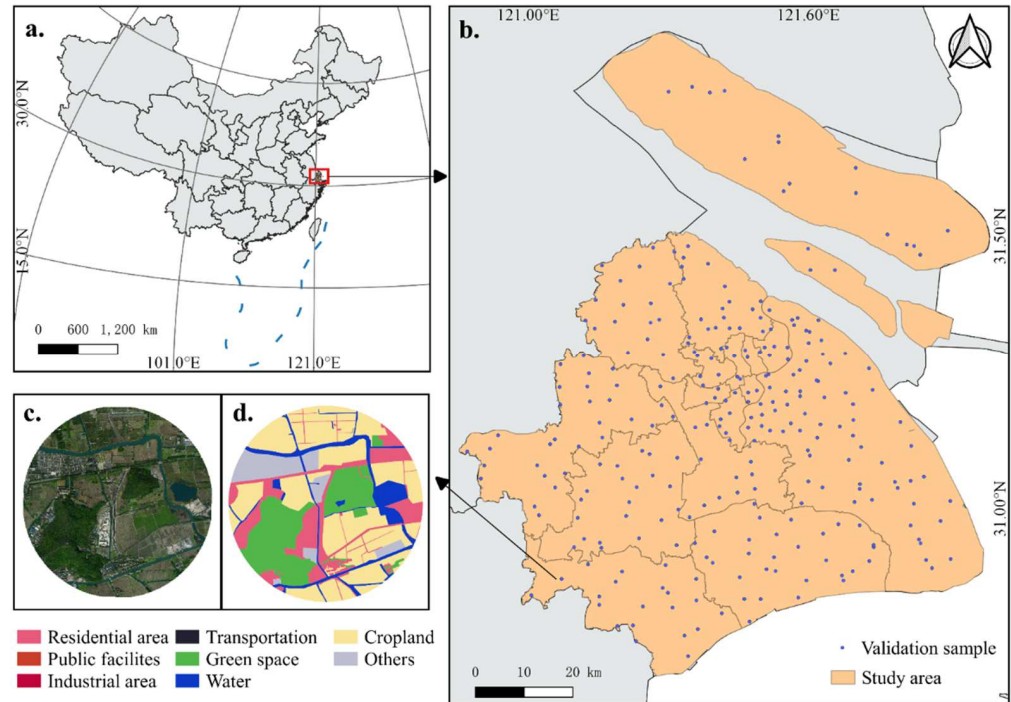

**Figure 4.** Randomly selected 1 km-radius circles in Shanghai for validation. (**a**) The location of Shanghai in China; (**b**) the spatial distribution of validation circles; (**c**) 0.5 m spatial resolution aerial photo of one sample circle; (**d**) land use land cover map of the circle in (**c**).

## 3. Results

### 3.1. Endmember Discrimination by Phenology Integration and Fisher Transformation

The endmember reflectance curve from the original image and their corresponding Fisher features for F-LSMA (which are extracted from winter image) (Figure 5a–c) and

PF-LSMA (which are extracted from the winter-summer composite image) (Figure 5d–f) are illustrated. The corresponding Fisher transformation weight coefficients are shown in Tables 2 and 3. As Figure 4 shows, the Fisher transformation can significantly reduce the within-endmember variability (Figure 5a,b,d,e). However, it still cannot effectively separate the soil and impervious surface with low albedo (Figure 5c). When integrating phenology information, the seasonally exposed soil endmembers were distinctly separated from low albedo in the Fisher feature space (Figure 5f). The phenology integration markedly enhanced the differentiation between the endmember spectral signatures of low albedo and seasonally exposed soil in blue, green, red, and SWIR2 band in the composite image (Figure 5d) compared with the winter image (Figure 5a). Moreover, the spectral signatures of seasonally exposed soil are distinctive from the evergreen vegetation in the band of SWIR1 in the composite image (Figure 5d). This will greatly enhance the discrimination of endmembers in the Fisher feature space (Figure 5f), and it will markedly improve the accuracy of extracting ISA.

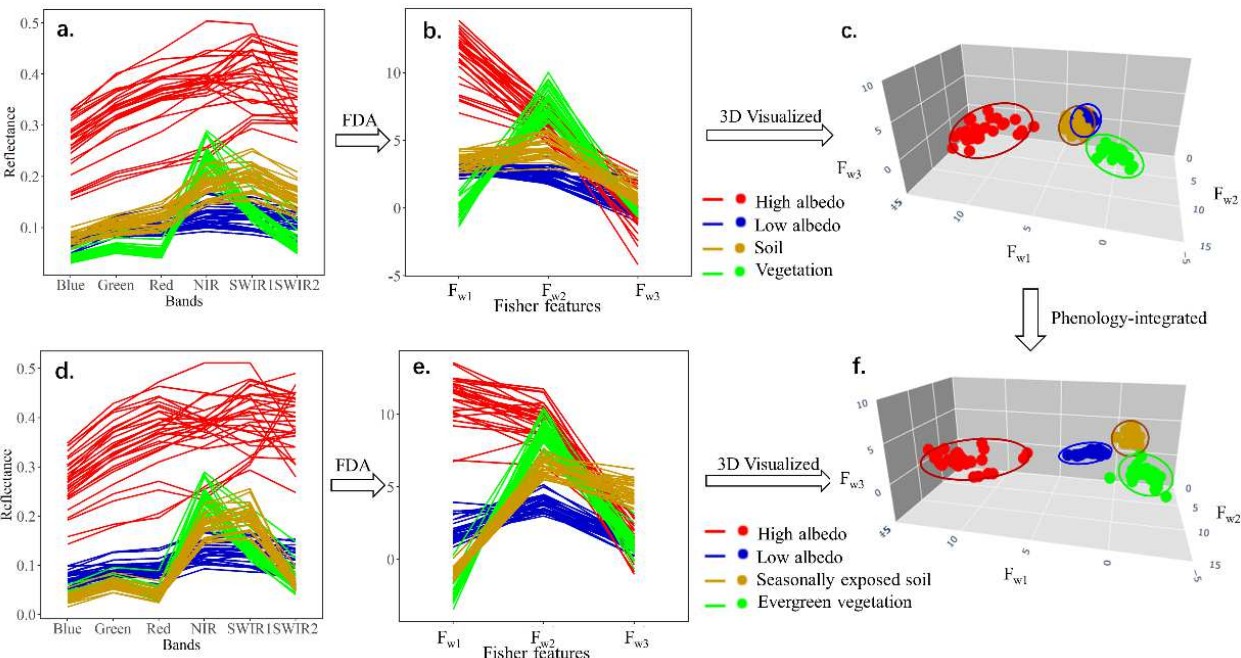

**Figure 5.** Endmember spectral signatures in surface reflectance and the corresponding Fisher transformation features, where (**a**–**c**) are the spectral reflectance, the Fisher features, and the distribution in Fisher feature space of endmember candidates for F-LSMA, respectively; (**d**–**f**) are the spectral reflectance, the Fisher features, and the distribution in Fisher feature space of endmember candidates for PF-LSMA, respectively.

**Table 2.** The weight matrix of Fisher linear discriminate analysis for endmember candidates in phenology-integrated imagery.

|  | $w_1$ | $w_2$ | $w_3$ |
| --- | --- | --- | --- |
| Blue (Summer image) | 0.006049 | −0.00178 | 0.006383 |
| Green (Summer image) | −0.00804 | 0.014469 | −0.01551 |
| Red (Summer image) | 0.00883 | −0.01271 | 0.009168 |
| NIR (Winter image) | −0.00344 | 0.002205 | 0.003375 |
| SWIR1 (Winter image) | −0.00038 | 0.000429 | −0.0046 |
| SWIR2 (Summer image) | 0.000697 | 0.001588 | 0.001412 |
| Proportion of trace | 0.8287 | 0.1601 | 0.0112 |

**Table 3.** The weight matrix of Fisher linear discriminate analysis for endmember candidates in single-date imagery.

|  | $w_1$ | $w_2$ | $w_3$ |
|---|---|---|---|
| Blue (Winter image) | 0.000997 | 0.003179 | 0.003115 |
| Green (Winter image) | 0.002031 | −0.00903 | 0.00438 |
| Red (Winter image) | 0.001651 | 0.00643 | −0.00301 |
| NIR (Winter image) | −0.00339 | −0.00252 | 0.000207 |
| SWIR1 (Winter image) | 0.000964 | 0.000778 | −0.0025 |
| SWIR2 (Winter image) | 0.000206 | −0.00172 | −0.00056 |
| Proportion of trace | 0.7117 | 0.2574 | 0.0309 |

### 3.2. Endmember Fractions and Validation

The unmixing result for Shanghai in 2015 quantifies the morphology for impervious surface, evergreen vegetation, and seasonally exposed soil. The result shows that highly urbanized areas were mainly concentrated in the city center and the surrounding towns with higher ISA fractions (Figure 6c). The evergreen vegetation is widely dispersed throughout the city. The seasonally exposed soil is mainly concentrated in the suburban, especially in the districts of Jinshan, Songjiang, and Qingpu (south-west), and Chongming island (north-east).

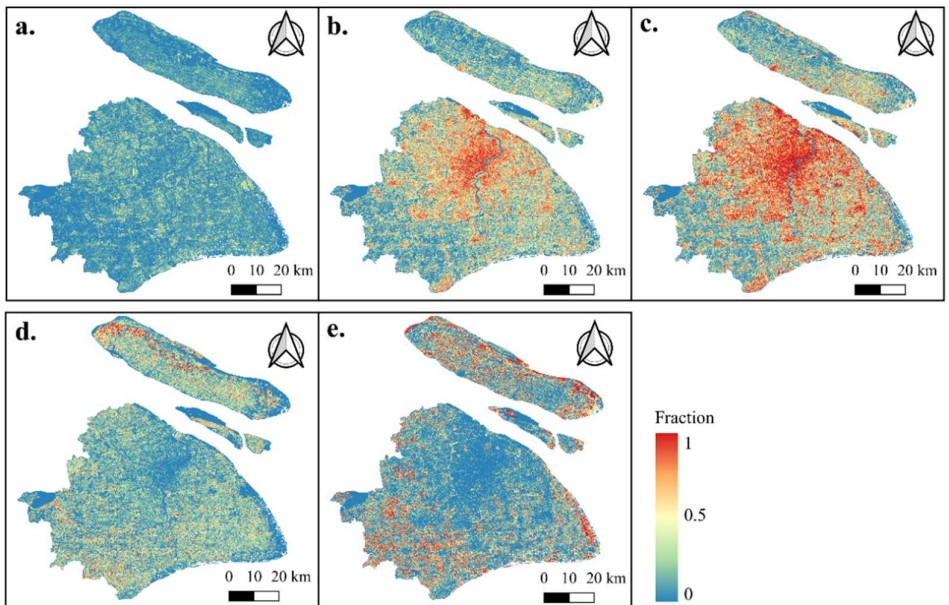

**Figure 6.** The endmember fraction map of high albedo (**a**), low albedo (**b**), impervious surface (**c**), evergreen vegetation (**d**), and seasonally exposed soil (**e**) in Shanghai in the year 2015.

The ISA products of PF-LSMA, F-LSMA, GAIA [5], and NUACI [63] were all validated using the reference data. Our results showed that the PF-LSMA performed the best among the ISA products compared with the reference. PF-LSMA has the highest accuracy in ISA estimation with the smallest RSME of 0.1112 and the highest R of 0.9483 (Figure 7a). The F-LSMA has a lower accuracy with RSME of 0.1327 and R value of 0.8835 (Figure 7b). The NUACI and GAIA methods have relatively lower estimation accuracy with RSME values of 0.1473 and 0.1429, and R value of 0.9108 and 0.9311, respectively (Figure 7c,d). All methods show some systematic errors to a certain extent.

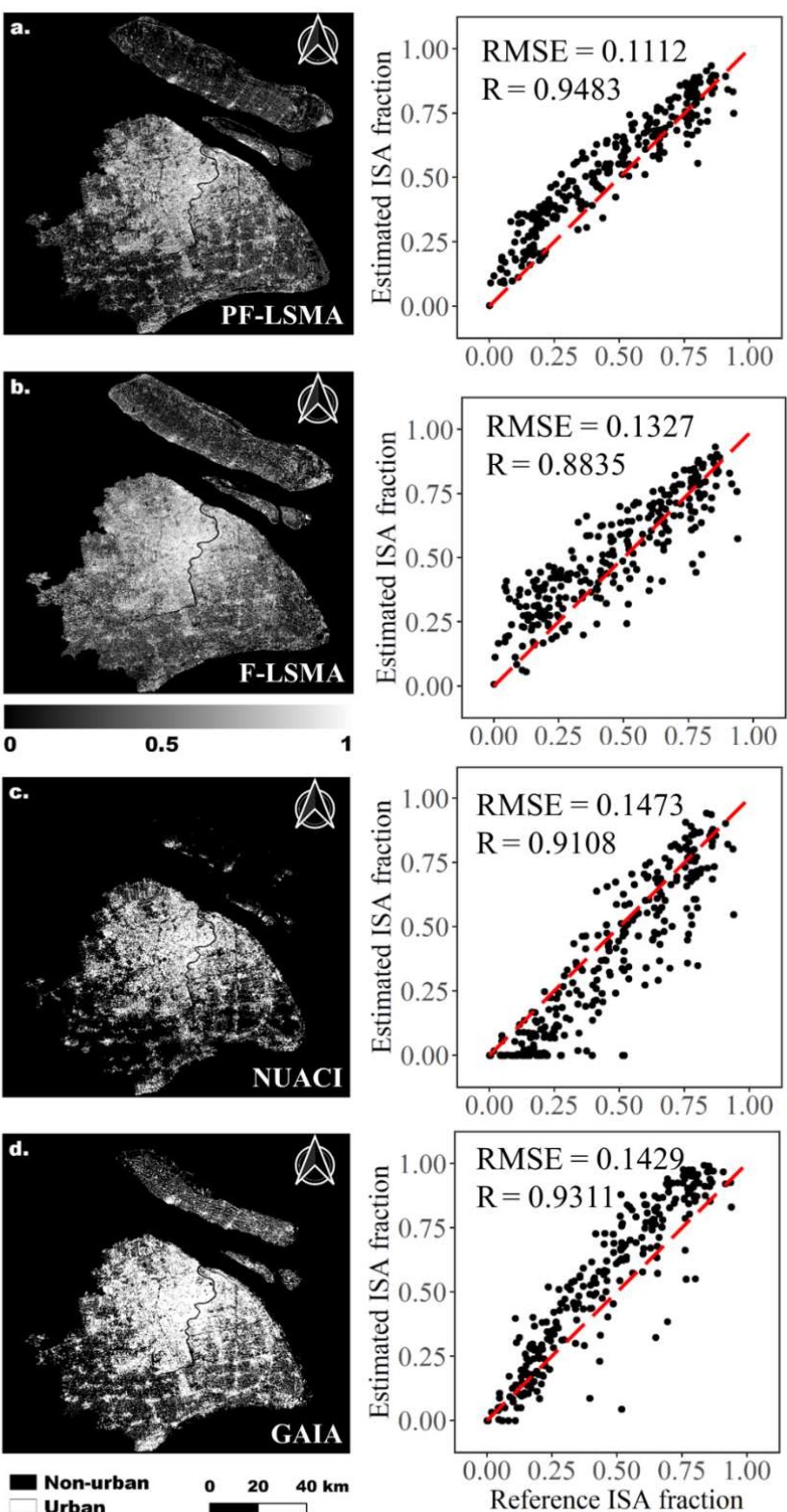

**Figure 7.** Comparison of the ISA fraction accuracy between different methods: PF-LSMA (**a**), F-LSMA (**b**), NUACI (**c**) [63], GAIA (**d**) [5]. The red line is the 1:1 line for reference.

## 4. Discussion

### 4.1. Phenology Combined with Fisher Transformation to Enhance the ISA Extraction

This study developed a new SMA approach, the PF-LSMA, for extracting ISA fractions at the subpixel level with improved accuracy compared with the approach without using vegetation phenology and the existing ISA datasets. A major challenge of using SMA to extract ISA under the framework of the V-I-S model is the within-endmember variations and between-endmember similarity in the spectral signature [26,27]. Data transformation, such as principal component analysis (PCA) [64], maximum noise fraction (MNF) transformation [25], and spectral normalization [24], is a useful approach to mitigate the problem, and consequently enhance the accuracy of ISA estimation. Jin, Wang, and Zhang [34] proposed a new method of SMA based on Fisher discrimination null space (FDNS), which is a linear transformation that reduces the within endmember spectral variability but enlarges between-endmember differences, to better decompose the mixed pixels in hyperspectral imagery with higher accuracy. Xu, Cao, Chen, and Somers [35] further integrated Fisher transformation in the multiple endmember spectral mixture analysis (MESMA) to effectively maximize the between-endmember variability and minimize the within-endmember variability and therefore obtain the highest accuracy of ISA estimation from Landsat images covering the five cities' urban area (the RSME is averagely 0.1346) compared with the other five spectral mixture analysis approaches. Our study showed that the Fisher transformation can successfully discriminate the endmember groups in the Fisher feature space. However, it cannot explicitly separate the impervious surface with low albedo and the seasonally exposed soil (Figure 5a–c). This may be due to the fact that the seasonally exposed soil manifests as a wet and dark surface in the warm and wet subtropical climate in Shanghai and is very similar to the impervious surface with low albedo in spectral signature. Therefore, it is difficult to separate them from each other. This is different from the situation in the arid region (e.g., Las Vegas), where bright soil can be successfully discriminated from the impervious surface with high albedo [35]. Our results show that although the Fisher transformation did not completely separate the seasonally exposed soil from the low albedo impervious surface in the Fisher feature space, it still produced satisfactory ISA outcomes when integrated into LSMA. The RSME and R value reached 0.1327 and 0.8835, respectively (Figure 7b), compared with the results from integrating Fisher transformation into MESMA [35] whose RSME and R values are 0.1346 and 0.8653, respectively.

Phenological information is also manifested to be useful to facilitate the ISA extraction [37,38,54,65,66]. Previous studies on SMA for ISA can be categorized into two types. One directly uses the phenological information, such as the high temporal resolution MODIS NDVI, the phenological metrics, to extract ISA. For instance, the temporal mixture analysis (TMA) technique, which used the rearranged MODIS NDVI time series dataset at the so-called temporal stable zone, i.e., from the first to the sixth largest NDVI values in a year, to estimate the ISA at city scale in Japan. The TMA-based method can largely reduce the effects of endmember variability and had a promising accuracy for estimating ISA with a lower RSME value of 8.7%. However, the TMA-based method cannot be applied to satellite data with a high spatial resolution which usually has low temporal resolution and would not provide appropriate temporal profiles for TMA [37]. Liu, Luo, and Yao [66] calculated nine phenological metrics (i.e., the start of the growing season, end of the growing season, length of the growing season, large seasonal integral, small seasonal integral, time of the peak growing season, etc.) from the seven-day interval EVI time series data fused from MODIS and Landsat EVI, as a series of discriminative features to be integrated in the Random Forest model to map ISA. Due to the indirect phenological information, this approach suffers from the urban landscape heterogeneity, vegetation composition, and spatial distribution, and it introduces a relatively large model uncertainty.

The other is using the surface reflectance of satellite imageries to include the phenological information. For example, Sung and Li [65] examined the plant phenology effect on the ISA mapping in a subtropical region and concluded that the winter image can generate sub-pixel impervious surface maps with the highest accuracy due to: firstly, the evergreen

plants maintain distinct spectral profile and green leaves helps separate impervious surface from vegetated cover; secondly, the impervious surface underneath a leave-off deciduous tree canopy can be more easily detectable in winter. Deng, Li, Zhu, Lin, and Xi [54] studied three areas under humid continental, tropical monsoon, and Mediterranean climates, and they demonstrated the importance of phenology in the ISA mapping. Firstly, they found that vegetation phenology influences the ISA mapping. In the temperate region with a humid continental climate, the performance of the summer image is better than the winter image. Because most of the vegetation in this climate region is senescent in the winter, the bare soil and tree trunk, whose spectral signatures are similar to the dark urban impervious surface, are exposed. During the summer, the bare soil and tree trunk are fully covered by green leaves, reducing spectral confusion. However, in regions with tropical monsoon or Mediterranean climates, the spring and summer images are better than the fall and winter images for ISA estimation. Secondly, they found that the multi-seasonal image composite can improve the accuracy of ISA extraction. The ISA estimation using multi-seasonal images was better than using the single-date image, and among all the image combinations, using the two seasonal images was the best [54]. Our results are similar to the results of Sung and Li [65] and Deng, Li, Zhu, Lin and Xi [54], and confirm that the two-image combination performs better than the single-date image. Particularly, our results demonstrate that the integration of Fisher transformation and phenology into the spectral mixture analysis can best discriminate the impervious surface, vegetation, and the seasonal bare soil (Figure 5f). However, plant phenology has complex effects on urban impervious surface extraction [65], whether it is positive or negative, and which seasonal image should be considered depends on the local context, such as climate, urban landscape heterogeneity, and vegetation composition.

### 4.2. Advantages and Limitation Compared with Other ISA Products

The impervious surface coverage is key information for urban planning and environmental management [6]. The percentage, magnitude, location, geometry, spatial pattern, and the perviousness–imperviousness ratio are important parameters for a range of issues and themes in urban ecological and environmental sciences [23]. Numerous studies have been conducted to extract the impervious surface from local to global scales. These studies generally can be divided into two categories, one used the various classification methods to extract or map ISA at the pixel level, and the product of ISA can cover from local, to, regional, and to global scales [5,17,67,68]; the other used the spectral unmixing method to extract ISA at the subpixel level [21,24,25,29,33,35,69]. Most studies have focused on the local scales.

Our results showed that the phenology-integrated and the Fisher transformed linear spectral mixture analysis (PF-LSMA) that we developed in this study can also capture and enhance the subpixel ISA estimation when we zoomed in some representative areas for further detailed comparisons of the ISA fractional map between different methods (Figure 8). The ISA map from PF-LSMA captured the spatial details for both high and low albedo impervious surfaces (Figure 8d,f). Moreover, PF-LSMA suppresses the tendency of the seasonally exposed soil to be identified as ISA (Figure 8e). Compared with F-LSMA, PF-LSMA showed better performance in discriminating impervious surface and seasonally exposed soil. In the ISA product of F-LSMA, more seasonally exposed soil was mistakenly identified as impervious surface (Figure 8h), while some of the impervious surfaces were not identified (Figure 8g,i). For example, in a high ISA area, e.g., Pudong Airport in Figure 8g, some ISA was mapped as soil by mistake when implementing F-LSMA. In contrast, the PF-LSMA showed better performance in the same area by taking advantage of the phenology difference between the winter and summer images.

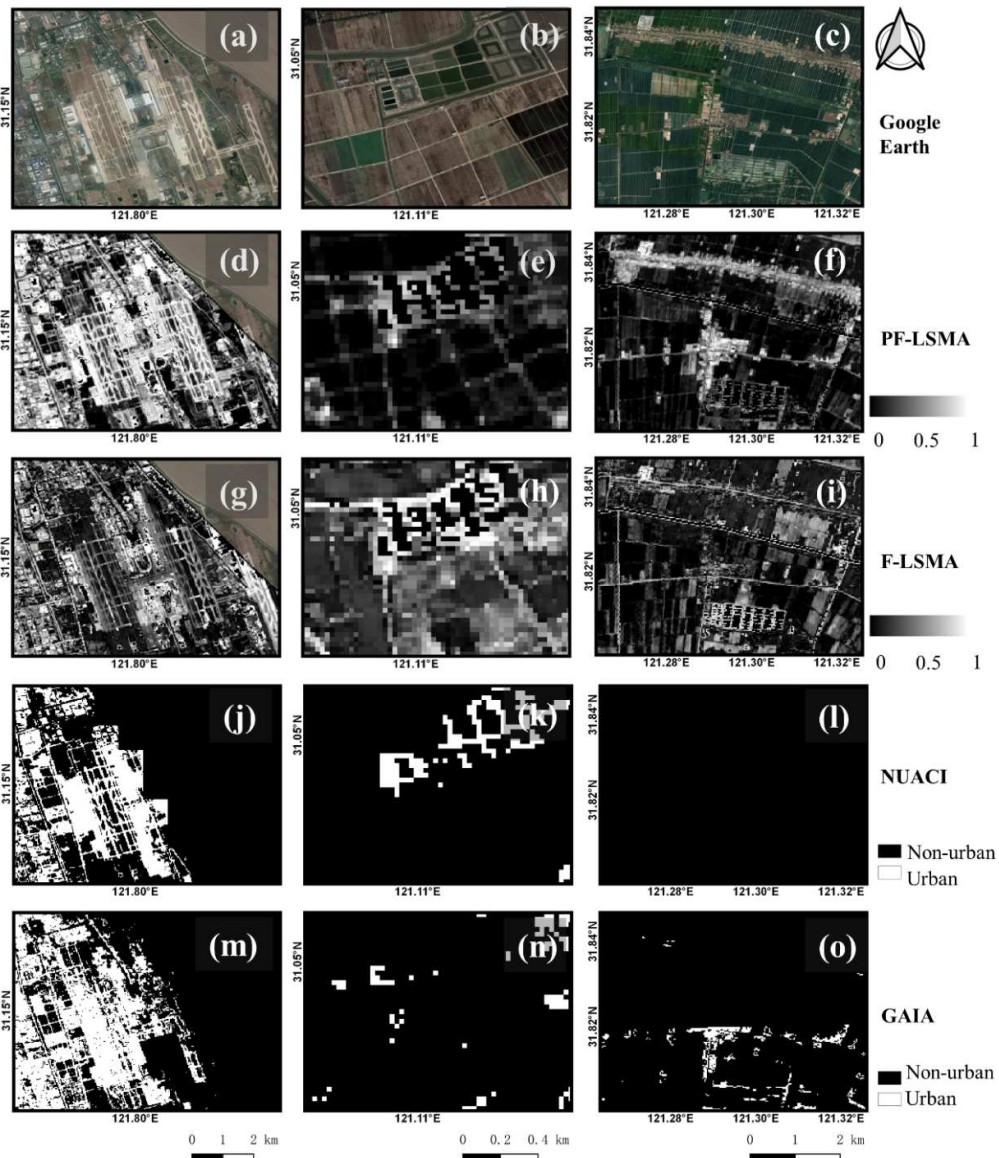

**Figure 8.** Comparisons of ISA extraction results from PF-LSMA, F-LSMA, NUACI [63], and GAIA [5]. The ISA result of Google Earth Engine (**a**–**c**), PF-LSMA (**d**–**f**), F-LSMA (**g**–**i**), NUACI (**j**–**l**), and GAIA (**m**–**o**) in 2015 in Shanghai.

Compared with other existing global products, PF-LSMA exhibited an improvement in the accuracy of ISA extraction and more spatial details of the ISA. Based on RMSE and R shown in Figure 7, it is evident that PF-LSMA shows higher accuracy in estimating the ISA fraction in Shanghai than the other three data products. Moreover, the spatial details of the ISA from PF-LSMA also agree with visual evaluation based on high-resolution images (Figure 8j–o). In contrast, the Normalized Urban Areas Composite Index (NUACI) [63] underestimated the ISA, while Global Artificial Impervious Areas (GAIA) [5] overestimated it. Both of them lost a lot of spatial details of the ISA (Figure 8j,l). The ISA underestimation by NUACI may be because: (1) The binary masking for "potential urban area" is generated from nighttime light data (DMSP-OLS); however, there is a substantial proportion of the urbanized area without or with weak night light, which may not be captured by DMSP-OSL. (2) NUACI identifies urban land by setting a threshold value which can omit a large number of mixed pixels with fractional ISA. The spatial variation in the threshold can result in unstable ISA estimation. The overestimation of ISA by GAIA may be due to (1) the definition of impervious surface. They used the "exclusion-inclusion" algorithm [16] which

defines human settlements as "dominated by the built environment with a proportion >50% within a pixel". The human settlements then were directly used to represent impervious surfaces in their study, thus largely overestimating the actual ISA fraction. (2) The one-way urban mask approach, which assumes that the transformation from urban to other land cover types is unlikely to happen [70]. This would result in the overestimation of ISA because the multi-directional urban land cover change, i.e., the urban land cover can be turned into green spaces, especially during the urban renewal processes [71]. Moreover, PF-LSMA is a sub-pixel classification that can provide the fractional information of every pixel. Thus, it is superior to those pixel-based or object-based classifications used in the NUACI [63] and GAIA [5] products, especially for moderate spatial resolution images (e.g., the Landsat series sensors) with a large number of mixed pixels in the highly heterogeneous urban area.

Although PF-LSMA has better performance in ISA extraction, it still has several limitations that can be improved in the future. First, as it shows in the validation section (Figure 7a), the result of PF-LSMA also has notable (even if it is relatively less) overestimation in the low ISA fractions (ISA fraction < 0.5) that is affected by the quality of the winter and summer images. Some of the seasonally exposed soil areas, which are mostly cropland in Shanghai, are in the after-harvest and not covered by vegetation in both summer and winter images, thus the phenological information cannot completely capture those areas. This results in overestimation in ISA by taking these seasonally exposed soil areas without expected phenological variation. Therefore, better incorporation of phenological information by considering vegetation composition into the multi-temporal image selection is further needed to improve the performance of PF-LSMA. Second, our High albedo-Low albedo-Evergreen Vegetation-Seasonally exposed soil endmember (H-L-EV-SS) model only considers seasonally exposed soil due to our study area locates in the subtropical monsoon climate region, only seasonally exposed soil could be found. More endmembers should be considered in other places. Third, our study was conducted in Shanghai with a subtropical climate; whether the model is suitable for other cities in different climate regions, where large areas of permanent soil or desert exist, still needs to be tested.

## 5. Conclusions

This study developed a new spectral mixture analysis model, the phenology-integrated and Fisher transformed linear spectral mixture analysis (PF-LSMA), to improve the accuracy of ISA fraction estimation from Landsat 8 OLI images. Vegetation phenology was creatively integrated in the SMA through the generation of a composite image from a pair of summer and winter images. The composite image consists of OLI bands 2–4, and 7 from the summer image and OLI bands 5 and 6 from the winter image. Phenology helps differentiate impervious surfaces and seasonally exposed soil, while the Fisher transformation reduces the within-endmember variability and enhances the between-endmember variability, hence, improving the accuracy of ISA extraction by enhancing the discrimination between endmembers. The PF-LSMA (RMSE = 0.1112, R = 0.9483) outperforms the algorithm without incorporating phenology, i.e., F-LSMA (RMSE = 0.1327, R = 0.8835), and the existing global ISA products (GAIA: RMSE = 0.1429, R = 0.9311; NUACI: RMSE = 0.1473, R = 0.9108). The PF-LSMA was developed on the Google Earth Engine platform, thus can be easily adapted for studies in other cities for further testing. To fully automate the process, an algorithm of automatic endmember selection is needed in the future.

**Author Contributions:** Conceptualization, J.L., C.S. and L.O.; Data curation, L.O., C.W., Y.L. and M.W.; Formal analysis, L.O.; Funding acquisition, J.L., C.W., and C.S.; Investigation, L.O., C.W., Y.L. and M.W.; Methodology, L.O. and C.W.; Project administration, J.L. and C.W.; Resources, J.L.; Software, L.O.; Supervision, J.L., J.H., C.S. and Q.Y.; Validation, L.O., Y.L. and M.W.; Visualization, L.O.; Writing—original draft, L.O.; Writing—review and editing, J.L., C.W., C.S., Q.Y., D.H. and J.H. All authors have read and agreed to the published version of the manuscript.

**Funding:** This research was funded by National Key R&D Program of China (No. 2017YFC0505801-01 to J. Li), the National Natural Science Foundation of China (No. 31870453 to J. Li, No. 32001162 to C. Wu, and No. 31528004 to C. Song).

**Data Availability Statement:** Data is available in USGS: https://www.usgs.gov/landsat-missions/landsat-surface-reflectance, accessed on 20 March 2022.

**Acknowledgments:** We are grateful to Shanghai Surveying and Mapping Institute for providing high spatial resolution images.

**Conflicts of Interest:** The authors declare no conflict of interest.

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
