# Peer review of "Mapping Impervious Surface Using Phenology-Integrated and Fisher Transformed Linear Spectral Mixture Analysis"

_remotesensing, doi:10.3390/rs14071673_

Round 1

Reviewer 1 Report

Dear authors,

I found your paper “Mapping impervious surface fraction using phenology-integrated and Fisher transformed linear spectral mixture analysis based on Google Earth Engine” interesting and the topic is important in the context of evaluating anthropogenic impact over natural ecosystems. However, I have a few recommendations to make.

The article is quite long and sometimes difficult to follow, although English is well written (minor spelling have to be revised). The methodological aspects are detailed, but it would be desirable to include in the appendix the scripts used in Google Earth Engine and R in order to facilitate the reproduction of your results or use in similar areas. Conclusions are too general and poorly supported by the results. You need to better emphasize the advantages of your methodology over similar work. Also, take into consideration changing the title to a shorter and simpler one.

Specific comments:

Under the “Materials and methods” chapter, study area. Please put the study area in a context map (country, region), so an international reader can easily spatially locate your case study. You can do this by using extending the data from Figure 4.

Line 173 replace “OIL” with “OLI”

Lines 193 - 194 please reconsider using terms “seasonal soil” and “permanent bare soil” with for example “seasonal vegetation cover” and respectively “bare soil”

Lines 204 – 206 “The phenology can be incorporated by composing the satellite images from two different seasons …” please provide some references here.  As phenology term means “the study of cyclic and seasonal natural phenomena, especially in relation to climate and plant and animal life” it may be that using only two different images acquired in the growing season and the dormant season would not refer to phenology but to vegetation types (eg. evergreen and deciduous).

Reviewer 2 Report

The study sought to develop a novel method that incorporates phenological information and Fisher transformation into the phenological information and Fisher transformation linear spectral mixture analysis (PF-LSMA)(PF-LSMA) to improve the accuracy of  impervious surface area (ISA) extraction. The authors hypothesized that directly integrating phenological information into  SMA by compositing the spectral features over seasons can enhance the separation of soil and impervious surface, hence, improving the accuracy of ISA extraction.

Generally, the study is interesting and and would be of interest to readers interested in urban land use land cover mapping. I however suggest that the manuscript be subjected to thorough English editing as the manuscript is littered with numerous grammatical errors; particularly the mixing of tenses. A few are listed below;

Line 37: "Lives" should be "lived"

38: "was" should be "is"

47: Delete "one"

59: Insert "that" after "assumes"

138: Should be "Shanghai is located in"

139: "was" should be "is"

163 and 166: Consistency in use of tenses - "have been" should be "were"

184: "is" should be "was"

213:Insert "all" after "where" and delete "all" after "endmembers"

214: "will be" should be "were"

220: Delete "one"

243: Delete "of them"

383: Delete "clear evidence"

422: Should be "This may be due to the fact that"

428: Replace "has" with "did"

429: Delete "has"

430: Delete "it is"

466: Review the use of "leave reaches"

470: "also" should come after "improve"

507: Should be "were not extracted"

Reviewer 3 Report

Dear authors,

Your paper is interesting, but the introduction need big improvement. You are using different terms, but this terms are not explain in the text and not all readers can be familiar and this can lead to lover interest. So please explain better in understandable form:

  • What you understand as Impervious Surface Area. It could be understood differently. Without explanation for some readers paper will not be clear.
  • Tell more about spectral mixture analysis
  • Introduce in better details SMA technique (SASM
  • Please explain, how you are using Phenology

Round 2

Reviewer 1 Report

Dear authors,

Thank you for using some of my suggestions, however, many things remained the same as in the previous version. 

I can find the GEE script but I did not find the R script using the MASS package mentioned in the methodology section at lines 272 - 273.

I still can't agree with the term "seasonal soil" as its meaning is not correct.  The soil itself does not have any seasonal character, only processes occurring in the soil layers could have seasonal behavior (eg. microbial activities, nutrient cycles, etc.). In your manuscript, several definitions are used for this term Lines 192  - 194 "The seasonal soil refers to deciduous tree species and crops that both have distinct phenology patterns. The seasonal soil is covered by either crops or deciduous trees during the growing season but exposed as bare soil after harvest or during the leaf-off season." Line 327 "the seasonal soil manifests as bare soil in the winter image" or Line 382 "The seasonal soil, which is mostly cropland ...", from these definitions it can be easily inferred that we are talking about the seasonality of the land use land cover while the soil remains the same.

Reviewer 3 Report

Dear authors,

thanks for update. I think, that my concerns are explained well

Author Response

Dear reviewer,

Thank you for your constructive suggestions which are valuable to help improve our manuscript. We are so happy that our answers have explained all your concerns.